# A Comprehensive Review on Radiomics and Deep Learning for Nasopharyngeal Carcinoma Imaging

**DOI:** 10.3390/diagnostics11091523

**Published:** 2021-08-24

**Authors:** Song Li, Yu-Qin Deng, Zhi-Ling Zhu, Hong-Li Hua, Ze-Zhang Tao

**Affiliations:** 1Department of Otolaryngology-Head and Neck Surgery, Renmin Hospital of Wuhan University, 238 Jie-Fang Road, Wuhan 430060, China; 2019183020073@whu.edu.cn (S.L.); RM001651@whu.edu.cn (Y.-Q.D.); 2019283020238@whu.edu.cn (H.-L.H.); 2Department of Otolaryngology-Head and Neck Surgery, Tongji Hospital Affiliated to Tongji Medical College, Huazhong University of Science and Technology, Wuhan 430030, China; zhuzhiling1993@hotmail.com

**Keywords:** nasopharyngeal carcinoma, deep learning, radiomics, imaging

## Abstract

Nasopharyngeal carcinoma (NPC) is one of the most common malignant tumours of the head and neck, and improving the efficiency of its diagnosis and treatment strategies is an important goal. With the development of the combination of artificial intelligence (AI) technology and medical imaging in recent years, an increasing number of studies have been conducted on image analysis of NPC using AI tools, especially radiomics and artificial neural network methods. In this review, we present a comprehensive overview of NPC imaging research based on radiomics and deep learning. These studies depict a promising prospect for the diagnosis and treatment of NPC. The deficiencies of the current studies and the potential of radiomics and deep learning for NPC imaging are discussed. We conclude that future research should establish a large-scale labelled dataset of NPC images and that studies focused on screening for NPC using AI are necessary.

## 1. Introduction

Nasopharyngeal carcinoma (NPC) is an epithelial carcinoma arising from the nasopharyngeal mucosal lining [1]. According to data from the International Agency for Research on Cancer, the number of new cases of NPC in 2020 was 133,354, of which 46.9% were diagnosed in China, showing an extremely uneven geographical distribution [2] (Figure 1). Although NPC accounts for only 0.7% of all malignant tumours and is relatively rare compared with other cancers, it is one of the most common malignant tumours in head and neck cancer [2,3]. NPC occurs more often in males, and the incidence of NPC in males is 2.5 times higher than in females [4]. Heredity and genes play important roles in the development of NPC [5,6,7], and the Epstein–Barr virus infection is perhaps the most common causal agent [1,8]. According to the differentiation of tumour cells, the WHO classified NPC into three types in 2003: keratinizing squamous cell carcinoma, non-keratinizing carcinoma, and basaloid squamous cell carcinoma [9]. The prognosis of NPC is generally better than that of most other cancers, with the reported overall 5-year survival rate reaching 80% [10]. Radiotherapy for early NPC and concurrent chemoradiotherapy for advanced NPC are recommended by the National Comprehensive Cancer Network [11]. Optimum imaging is crucial for staging and radiotherapy planning for NPC [12]. There are various general image inspections for NPC, including computed tomography (CT), magnetic resonance imaging (MRI), and electronic endoscopy. Compared with CT, MRI is the preferred method for primary tumour delineation because of its high resolution on soft tissue [13,14].

In recent years, artificial intelligence (AI) has been rapidly integrated into the field of medicine, especially into medical imaging. Research on the application of AI in NPCs has also gradually become a hot topic. Lancet has published a series of reviews titled ‘Nasopharyngeal carcinoma’ every few years [1,12,15,16]. In the most recent review published in 2019, 18 research questions on NPC that remain to be answered were proposed, and two of them were about AI and NPC: ‘How can reliable radiomics models for improving decision support in NPC be developed?’ and ‘How can artificial intelligence automation for NPC treatment decisions (radiotherapy planning, chemotherapy timing, and regimens) be applied?’. Subsequently, many articles in this area have emerged, and a large number of studies have reported on tumour detection, image segmentation, prognosis prediction, and chemotherapy efficacy prediction in NPC. In these studies, radiomics and deep learning (DL) have gradually become the most important AI tools.

In this work, we focus on the studies of radiomics and DL in the image analysis of NPC and aim to spread the implementation pipeline of radiomics and DL and discover the future potential of radiomics and DL in this field. Because each model in these studies is trained based on a different database which has a huge impact on the model, and because there are no indicators or validation protocols of consensus for the evaluation of each model’s performance, a comprehensive overview is presented to provide a holistic profile instead of a meta-analysis.

This paper will be presented in the following sections:

The pipeline of radiomics and the principle of DL are briefly described;The studies of radiomics and DL for NPC imaging in recent years are summarized;The deficiencies of current studies and the potential of radiomics and DL for NPC imaging are discussed.

## 2. Pipeline of Radiomics

The suffix-omics has initially arisen from ‘genomics’, which is generally defined as the study of whole genomes [17]. Radiomics, which was first proposed by Lambin in 2012 [18], is a relatively ‘young’ concept and is considered a natural extension of computer-aided diagnosis and detection systems [19]. It converts imaging data into a high-dimensional mineable feature space using a large number of automatically extracted data-characterization algorithms to reveal tumour features that may not be recognized by the naked eye and to quantitatively describe the tumour phenotype [20,21,22,23]. These extracted features are called radiomic features and include first-order statistics features, intensity histograms, shape- and size-based features, texture-based features, and wavelet features [24]. Conceptually speaking, radiomics belongs to the field of machine learning, although human participation is needed. The basic hypothesis of radiomics is that the constructed descriptive model (based on medical imaging data, sometimes supplemented by biological and/or medical data) can provide predictions of prognosis or diagnosis [25]. A radiomics study can be structured in five steps: data acquisition and pre-processing, tumour segmentation, feature extraction, feature selection, and modelling [26,27] (Figure 2). In some other reviews, the radiomics pipeline is described in four steps [25,28], in which feature selection and model building are grouped into one step, considering that the two steps are completed in a sequence in the program.

The collection and pre-processing of medical images is the first step in the implementation of radiomics. Radiomics relies on well-annotated medical images and clinical data to build target models. CT was first used when radiomics was proposed [18]. Subsequently, MRI [29], positron emission tomography (PET/CT) [30], and ultrasound [31,32] have been widely used for image analysis of different tumours using radiomics. Image pre-processing (filtering and intensity discretization) is essential as these images often come from different hospitals or medical centres, which results in differences in image parameters, and such differences may have unexpected effects on the model.

Image segmentation is a distinctive feature of radiomics. The methods of image segmentation generally include manual segmentation and semi-automatic segmentation [33]. The region of image segmentation determines which voxels will be included in the analysis, so image segmentation is a basic step of radiomics. However, there is no gold standard for image segmentation. The operation of different personnel will inevitably lead to differences of segmentation regions, which results in the heterogeneity of extracted image features and affects the performance of the model. The process of image segmentation is a very cumbersome step that depends on professionals, which leads to difficulties in obtaining high-quality data and increases the difficulty of clinical translation [34,35].

Feature extraction is a technical step in the pipeline of radiomics, which is implemented in software such as MATLAB. The essence of radiomics is to extract high-throughput features that connect medical images and clinical endpoints from images. These details must be included in the article as the process of feature extraction is affected by the algorithm, methodology, and software parameter setting [36,37]. The current radiomics pipeline typically incorporates approximately 50–5000 quantitative features, and this number is expected to increase [28].

The purpose of feature screening is to reduce the dimensionality of features and screen out the features most relevant to clinical endpoints to avoid overfitting of the model. When the feature dimension exceeds a limit, the performance of the classifier decreases with an increase in the feature dimension, and the time cost of training the model increases. Therefore, the selection of more effective feature subsets through feature selection algorithms is very important for establishing the model. According to the form of feature selection, the current feature screening methods are divided into three categories: filter, wrapper, and embedding [38]. Among them, the least absolute shrinkage and selection operator (LASSO), which is an embedding method, is the most commonly used [39,40].

The process of modelling entails finding the best algorithm to link the selected image features with the clinical endpoints. Supervised and unsupervised learning are common strategies [25,41]. The modelling strategy has been proven to affect the performance of the model [42]. Therefore, it is necessary to select the most appropriate algorithm according to the type of data and target. In addition, building multiple models simultaneously is a worthy, but not a necessary method. Model verification is an indispensable step in establishing the model. The best strategy is to use independent, external data to verify the performance of the model; this has not been implemented in many studies.

To date, radiomics has made impressive performances in tumour differentiation [43,44,45], prognosis evaluation [46,47,48], therapeutic effect evaluation [49,50,51,52], and tumour metastasis evaluation [53,54,55]. Compared with the performance of traditional predictive models based on clinical data and imaging anatomy, better performance of radiomics has been widely reported [24,56,57].

## 3. The Principle of DL

For a better understanding of DL, it is necessary to clarify the two terms of AI and machine learning, which are often accompanied by and confused with DL [58] (Figure 3). The concept of AI was first proposed by John McCarthy, who defined it as the science and engineering of intelligent machines [59]. In 1956, the AI field was first formed in a Dartmouth College seminar [60]. Currently, the content of AI has become much richer to include knowledge representation, natural language processing, visual perception, automatic reasoning, machine learning, intelligent robots, automatic programming, etc. The term AI has become an umbrella term [61]. Machine learning is a technology used to realize AI. Its core idea is to use algorithms to parse and learn from data, then make decisions and predictions about events in the real world [62], which is different from traditional software programs that are hard-coded to solve specific tasks [63]. The algorithm categories include supervised learning algorithms, such as classification and regression methods [64], unsupervised learning algorithms, such as cluster analysis [65], and semi-supervised learning algorithms [66]. DL is an algorithm tool for machine learning [67]. It is derived from an artificial neural network (ANN), which simulates the mode of human brain processing information [68], and uses the gradient descent method and back-propagation algorithm to automatically correct its own parameters, making the network fit the data better [69,70]. Compared with the traditional ANN, DL has more powerful fitting capabilities owing to more neuron levels [71]. According to different scenarios, DL includes a variety of neural network models, such as convolutional neural networks (CNNs) with powerful image processing capabilities [72], recurrent neural networks (RNNs), which primarily process time-series samples [73,74], and deep belief networks (DBNs), which can deeply express the training data [75]. In recent years, CNN-based methods have gained popularity in the medical image analysis domain [67,76,77]. In the studies of NPC imaging using DL models, CNN was adopted in almost all studies.

There are four key ideas behind CNNs that take advantage of the properties of natural signals: local connections, shared weights, pooling, and the use of many layers [68]. The structure of a CNN, which is mainly composed of an input layer, hidden layer, and output layer, is shown in Figure 4. The hidden layers consist of a convolutional layer, pooling layer, and a fully connected layer. After inputting an image, a greyscale image is converted into a single-channel matrix according to the grey value of each pixel, whereas a colour image is converted into a three-channel matrix. Fixed-size convolution kernels (usually 3 × 3) are used to sequentially scan the matrix area of the same size on the image, and the values of the convolution kernels are multiplied by the values of the corresponding position on the image matrix and finally summed. Each time the convolution kernel moves to the right according to a fixed step (after reaching the right edge of the image, it moves down one step and returns to the left edge of the image) to obtain a summed value. When the convolution kernels finish scanning the entire image, a new matrix, called a convoluted feature, is developed. The pooling layer is a process of downsampling the spatial dimension, with the aim of feature reduction, compressing the parameter number to reduce overfitting [78]. There are generally three methods of pooling: stochastic pooling, average pooling, and the most commonly used, max pooling. The most common size of a pooling filter is 2 × 2, and the most common step size is 2. Generally, the convolution and pooling processes of a CNN model are repeated many times, and the ‘depth’ of a DL model is embodied in the number of convolutions. Common CNN models are endowed with high depth and a large number of parameters. For example, the VGG-16 model, which won the runner-up in the 2014 ILSVRC competition, has a total of 16 layers with 138 million parameters. Usually, there are several fully connected layers at the bottom of a CNN. The fully connected layer maps the learned distributed feature representation to the sample label space and transforms the quantitative value into a nonlinear value through an activation function, which plays the role of a classifier in the model [72]. Owing to its principle, CNN has an advantage in processing image-related tasks and is widely used in medical imaging-related research [79].

Because of the differences in the principles behind deep learning and radiomics, there are differences in the specific tasks and advantages of their implementation processes. Because implementations of radiomics require manual segmentation of lesion areas to capture the radiomics features, this approach is more often used to perform the tasks of diagnosis prediction, assessment of tumour metastasis, and prediction of therapeutic effect. Deep learning models are often based on the whole image, which contains information on the relationship between the tumour and the surrounding tissues. Therefore, image synthesis, lesion detection, prognosis prediction, and image segmentation are regarded more commonly as tasks suitable for deep learning methods. Because the input image of most deep learning tasks is often a full image, which contains the noise information around the lesion, the performance of deep learning models is thus far not as good as that of radiomics for the same dataset due to the embedded noise information. However, because radiomics retains the fundamental disadvantage that manually defining the area of interest is strictly required, which necessitates the performance of considerable human labour and this is not required by deep learning methods, the datasets available for deep learning tasks could be much larger than those of the radiomics task. In addition, with the rapid development of deep neural network algorithms, the performance of deep learning is gradually improving and its performance in many tasks now exceeds the performance of radiomics. For example, Google’s EfficientNet series of networks, published in ICML 2019 [80], demonstrated an extremely impressive performance in terms of efficiency and accuracy of ImageNet tasks.

Although radiomics arose in 2012, and DL has captured the researchers’ vision since 2015, there have been few studies on NPC imaging before 2017. Therefore, most of the studies included in this paper are from January 2017 to March 2021. From the perspective of technology, radiomics and DL tasks in medical images include detection, segmentation, and most commonly, classification. However, these concepts are abstract for most clinicians who do not grasp the idea of AI. Therefore, we summarize the published studies in this area from the perspective of specific clinical issues to be solved.

After screening and systematically summarizing the retrieved literature, we summarized the specific issues that have been considered and divided the studies into three sections according to whether the studies adopted radiomics, DL, or both. Specific tasks using radiomics are divided into the following categories: prognosis prediction, assessment of tumour metastasis, tumour diagnosis, prediction of therapeutic effect, and prediction of complications. Specific tasks using DL are divided into prognosis prediction, image synthesis, detection and/or diagnosis, and image segmentation. We summarize the contributions, methods, and results of each paper in accordance with the chronological order of publication. To evaluate the model, we selected representative outcome indicators for concise presentation (external validation, best model, evaluation indicators of area under the curve (AUC), C-index, or Dice similarity coefficient (DSC) were preferred). A consolidated description of similar studies has been adopted.

## 4. Screening of Studies

Because there are no indicators or validation protocols of consensus for the evaluation of each model’s performance, and we firmly believe that comparison of models is tenuous for the heterogeneity of a database, a holistic profile of this field was provided instead of a meta-analysis. From this perspective, loose inclusion and exclusion criteria were set (Table 1). Finally, a total of 80 studies were included after following the inclusion and exclusion criteria (Figure 5).

## 5. Studies Based on Radiomics

### 5.1. Prognosis Prediction

Prognosis prediction includes tumour risk stratification and recurrence/progression prediction. Among the 31 radiomics-based studies retrieved, 17 were on this topic (Table 2).

#### 5.1.1. 2017

Zhang was one of the first researchers to apply radiomics to NPC imaging. In 2017, he published three studies in this area, all of which were based on MRI images. In [24], a risk stratification prediction model with a C-index of 0.776 was established using a nomogram. In [81], four risk stratification prediction models were established based on random forest, random forest and adaptive boosting, sure independence screening, and linear support vector machine. The AUC was used to evaluate the performance of the model. The best AUC for the validation cohort was 0.846. In [82], a prediction model for predicting the progression of an advanced NPC was established. The dataset included 113 patients. The tumour progression and outcome of patients were predicted according to the radiomics score of the model, and the AUC value reached 0.886.

In the study by Ouyang [83], 100 patients with advanced NPC were included. Radiomic features were extracted, and a Cox proportional hazards regression model was established based on MRI images. The model successfully stratified patients into low- or high-risk groups in the validation set (hazard ratio [HR] = 7.28, *p* = 0.015).

#### 5.1.2. 2018

Retrieved none.

#### 5.1.3. 2019

The study of Lv [84] is the only positron emission tomography (PET)/CT-based study, which was different from other studies in 2019. A total of 128 patients with NPC were included and 3276 radiomic features were extracted. Then, 13 clinical characteristics were selected in the study to establish seven predictive models using the Cox proportional hazard regression. The C-index was used to evaluate the performance of the models, and the best C-index value in the validation cohort was 0.77.

Several other MRI-based studies were conducted in 2019 [85,86,87,88,89,90,91], among which [85,91] used a support vector machine (SVM) to establish a prediction model after feature extraction and selection; the best C-index in the validation cohort of [85] was 0.814, while the AUC in [91] was 0.80. The Cox proportional hazard regression and nomogram were used to build predictive models in [86,87,89]; 737 patients were included in [86], and the C-index of the external validation cohort was 0.73, which was better than that of clinical prognostic variables (0.577, 0.605). In [87], 260 radiomic features were extracted from the primary tumour and lymph nodes on axial MRI, and LASSO was applied for feature selection and data dimension reduction. Finally, a C-index of 0.811 was obtained. In [89], univariate and multivariate analyses were used for feature selection from the 970 features that were extracted from 140 patients with NPC, and a radiomic nomogram was built by multivariate analyses, which finally reached a C-index of 0.74. In [90], clinical features of tumour volume, T stage, N stage, overall stage, age, and gender were added after extracting the imaging features. Then, the Cox proportional hazard regression analysis was used to determine the independent predictors of progression-free survival and establish a prediction model. The optimal AUC for the model was 0.825. In the study of [88], conventional imaging methods were used, and disease-free survival and overall survival were used as clinical endpoints. Finally, C-indices of 0.751 and 0.845, respectively, were obtained.

#### 5.1.4. 2020

In [92], a total of 128 patients were included, and PET/CT was used to build the dataset. The tumour of each patient was partitioned into several phenotypically consistent sub-regions based on individual- and population-level clustering. Subsequently, 202 features were extracted in each sub-region, and the imaging biomarkers and clinicopathological factors were evaluated using multiple Cox regression analyses and Spearman correlation analysis. It was concluded that the predictive performance of the biomarkers in the sub-regions with three PET/CT imaging characteristics is better than the predictive performance of the entire tumour (C-index, 0.69 vs. 0.58).

In [93,94], MRI images were used to build datasets that included 327 and 136 patients with NPC, respectively. LASSO and recursive feature elimination were used to select features in [93]. The author constructed five models to predict progression-free survival using the univariate Cox proportional hazard model, and the best C-index in the validation set was 0.874. A total of 530 stable features were extracted, and 67 non-redundant features were selected in [94]. Four predictive models were constructed based on the Cox proportional hazard model, and the C-index of the best model was 0.72.

In [95], 100 consecutive cases of NPC were recruited, and nine of the most relevant radiomic features were selected from features extracted from PET and MRI using LASSO. A predictive model of NPC staging was established based on logistic regression, and the total AUC in PET and MR were 0.84 and 0.85, respectively.

#### 5.1.5. 2021

In [96], a multiple model combined with SVM based on the PET/CT of 85 patients with stage III-IVB NPC was established. The model predicted local recurrence and distant metastasis of tumours with sequential floating forward selection and achieved an AUC of 0.829.

### 5.2. Assessment of Tumour Metastasis

#### 5.2.1. 2017–2018

Retrieved none.

#### 5.2.2. 2019

A classic radiomics approach was implemented in [97]. After extracting 2780 features from the MRI of 176 patients with NPC, LASSO was used for feature screening, and a radiomics model for predicting the distant metastasis of tumours based on logistic regression was established. The AUC for the validation set was 0.792.

#### 5.2.3. 2020

The authors in [98] developed an MRI-based radiomics nomogram for the differential diagnosis of cervical spine lesions and metastasis after radiotherapy. A total of 279 radiomic features were extracted from the enhanced T1-weighted MRI, and eight radiomic features were selected using LASSO to establish a classifier model that obtained an AUC of 0.72 with the validation set.

In [99], the authors explored the issue of whether there was a difference between radiomic features derived from recurrent and non-recurrent regions within the tumour. Seven histogram features and 40 texture features were extracted from the MRI images of 14 patients with T4NxM0 NPC. The author proposed that there were seven features that were significantly different between the recurrent and non-recurrent regions.

Manual delineation of the region of interest (ROI), which is widely used in current radiomics-related studies, has a high degree of variability. However, the tolerance of delineation differences and the possible influence of each step of radiomic analysis are not clear. In [100], based on 238 cases of NPC and 146 cases of breast cancer images, the author established a model for assessing sentinel lymph node metastasis by using a random forest algorithm and implementing erosion, smoothing, and dilation on the ROI. It was proposed that differences from smooth delineation or expansion with 3 pixels width around the tumours or lesions was acceptable.

#### 5.2.4. 2021

In 2021, the study of [96], which was introduced in the section on prognosis prediction, established a model for the assessment of tumour metastasis simultaneously. The best AUC for predicting tumour metastasis was 0.829 (Table 3).

### 5.3. Tumour Diagnosis

#### 5.3.1. 2017

Retrieved none.

#### 5.3.2. 2018

Lv [101] established a diagnostic model to distinguish NPC from chronic nasopharyngitis using the logistic regression of leave-one-out cross-validation method. A total of 57 radiological features were extracted from the PET/CT of 106 patients, and AUCs between 0.81 and 0.89 were reported.

#### 5.3.3. 2019

Retrieved none.

#### 5.3.4. 2020

In [102], 76 patients were enrolled, including 41 with local recurrence and 35 with inflammation, as confirmed by pathology. A total of 487 radiomic features were extracted from the PET images. The performance was investigated for 42 cross-combinations derived from six feature selection methods and seven classifiers. The authors concluded that diagnostic models based on radiomic features showed higher AUCs (0.867–0.892) than traditional clinical indicators (AUC = 0.817) (Table 4).

### 5.4. Prediction of Therapeutic Effect

#### 5.4.1. 2017

Retrieved none.

#### 5.4.2. 2018

Wang [103] established an MRI-based imaging omics model for the pre-treatment prediction of early response to induction chemotherapy. A total of 120 patients with stage II-IV NPC were enrolled, and the best AUC of the model was 0.822.

#### 5.4.3. 2019

Yu [104] established a radiomics model based on MR images to predict adaptive radiotherapy eligibility of patients with NPC before the start of the treatment in their study. After feature extraction, a double cross-validation approach of 100 resampled iterations with 3-fold nested cross-validation was employed in the LASSO logistic region for feature selection. Then, a prediction model was established, in which the method of modelling was not declared. An average AUC of 0.852 was reached with the testing set.

#### 5.4.4. 2020

In [105], 108 patients with advanced NPC were included to establish the dataset. The ANOVA/Mann–Whitney U test, correlation analysis, and LASSO were used to select texture features, and multivariate logistic regression was used to establish a predictive model for the early response to neoadjuvant chemotherapy. Finally, an AUC of 0.905 was obtained for the validation cohort.

In [106], logistic regression was used to establish a prediction model based on MRI images to predict the response of advanced NPC to the induction chemotherapy of a gemcitabine plus cisplatin (GP) regimen and docetaxel plus cisplatin (TP) regimen. In the validation cohort, the predictive ability of the established model for the GP regimen reached an AUC of 0.886, while the AUC in the TP regimen was 0.863.

In [107], 19 radiomic features were screened out by using t-test, LASSO regression, and leave-one-out cross-validation after feature extraction from 123 patients with NPC. These 19 radiomic features were combined with clinical features to establish a prediction model of induction chemotherapy based on SVM, which reached an AUC of 0.863 (Table 5).

### 5.5. Predicting Complications

#### 5.5.1. 2017–2018

Retrieved none.

#### 5.5.2. 2019

In [108], a radiomics model for predicting early acute xerostomia during radiation therapy was established based on CT images. Ridge CV and recursive feature elimination were used for feature selection, whereas linear regression was used for modelling. However, the study’s test cohort included only four patients with NPC and lacked sufficient evidence, despite the study reaching a precision of 0.922.

#### 5.5.3. 2020

The authors in [109] established three radiomics models for the early diagnosis of radiation-induced temporal lobe injury based on the MRIs of 242 patients with NPC. The feature selection in the study was achieved by the Relief algorithm, which is different from other studies. The random forest algorithm was used to establish three early diagnosis models. The AUCs of the models in the test cohort were 0.830, 0.773, and 0.716, respectively (Table 6).

## 6. Studies Based on DL

### 6.1. Prognosis Prediction

#### 6.1.1. 2017–2018

Retrieved none.

#### 6.1.2. 2019

In [110], a prognostic model based on 3D DenseNet, which is a convolutional neural network, to predict disease-free survival in 1636 patients with non-metastatic NPC was established. The model classified patients into low- and high-risk groups based on the cut-off value of risk scores, and the author claimed that the model could distinguish the two groups of patients correctly (HR = 0.62, *p* < 0.0001). Similarly, Du [111] established a deep convolutional neural network model for the risk assessment of patients with non-metastatic NPC. This study included 596 patients with NPC. The model achieved an AUC of 0.828 in the validation set for 3-year disease progression. However, it did not generalize well for the test set (AUC = 0.69), which consisted of 146 patients from a different centre.

#### 6.1.3. 2020

Yang [112] established a weakly-supervised, deep-learning network using an improved residual network (ResNet) with three input channels to achieve automated T staging of NPC. The images of multiple tumour layers of patients were labelled uniformly. The model output a predicted T-score for each slice and then selected the highest T-score slice for each patient to retrain the model to update the network weights. The accuracy of the model in the validation set was 75.59%, and the AUC was 0.943.

In [113], an end-to-end, multi-modality deep survival network was proposed to predict the risk of tumour progression and was compared with the traditional four popular state-of-the-art survival methods. Finally, the established multi-modality deep survival network showed the best performance, with a C-index of 0.651. Similarly, Cui [114] established several prognostic models of NPC based on DL and several other conventional algorithms, such as the generalized linear model, extreme random tree, gradient boosting machine, and random forest. The average AUCs for overall survival, distant metastasis-free survival, and local-region relapse-free survival results obtained from the image data-based model were 0.796, 0.752, and 0.721, respectively.

Qiang [34] established a 3-D convolutional neural network-based prognosis prediction system for locally advanced NPC using MR images and clinical data. The study included 3444 cases, which was one of only two studies that included a sample size of more than 2000. The C-index of the established network in the internal validation cohort and the three external validation cohorts reached 0.776, 0.757, 0.719, and 0.746, respectively.

In contrast to the previous study, Liu’s [115] model for predicting the prognostic value of individual induction chemotherapy based on the DeepSurv neural network used pathological images from 1055 patients. The established DL model (C-index: 0.723) performed better than the EBV DNA (C-index: 0.612) copies and the N stage (C-index: 0.593).

#### 6.1.4. 2021

In [116], a DL model based on ResNet was established to predict the distant metastasis-free survival of locally advanced NPC. In contrast to the studies published in 2020, the authors of this study removed the background noise and segmented the tumour region as the input image of the DL network. Finally, the optimal AUC of the multiple models combined with the clinical features was 0.808 (Table 7).

### 6.2. Image Synthesis

#### 6.2.1. 2017–2018

Retrieved none.

#### 6.2.2. 2019

In [117], Li used a deep convolutional neural network (DCNN) to generate a composite CT image based on cone-beam CT. The 1%/1 mm gamma pass rate of synthetic CT was 95.5% ± 1.6%. The author proposed that the DCNN model can generate high-quality synthetic CT images from cone-beam CT images and can be used to calculate radiotherapy doses for patients with NPC. Similarly, Wang [118] used a DCNN to generate CT images based on T2-weighted MRI. Compared with real CT, synthetic CT could accurately reconstruct most soft tissue and bone areas, except for the interface between soft tissue and bone and the interface between fragile structures in the nasal cavity.

#### 6.2.3. 2020

Tie [119] used a multichannel multipath conditional generative adversarial network to generate CT images from an MRI. The network was developed based on a 5-level residual U-Net with an independent feature extraction network. The highest structural similarity index of the network was 0.92.

In [120], a generative adversarial network was used to generate CT images based on MRIs to guide the planning of radiotherapy for NPC. The 2%/2 mm gamma passing rates of the generated CT images reached 98.68% (Table 8).

### 6.3. Detection and/or Diagnosis

#### 6.3.1. 2017

Retrieved none.

#### 6.3.2. 2018

There were three studies that focused on using neural networks based on nasal endoscopic images to detect and/or diagnose NPC in 2018, and two of them were the work of Mohammed [121,122]: an artificial neural network based on the Haar feature fear and genetic algorithm were used to establish an endoscopic diagnosis model for NPC. The authors included a total of 381 NPC endoscopic images, including 159 tumours (abnormal cases) and 222 normal tissues. The established network had a high precision of 96.22%, sensitivity of 95.35%, and specificity of 94.55%. Mohammed also established three different neural network models in another article [122]. The accuracies of the models reached 95.66%, 93.87%, and 94.82%. SVM, the k-nearest neighbour algorithm, and ANN were used in another study to identify NPC that seemed to be based on a coincident dataset with the other two articles [123]. Li’s study included 28,966 eligible images, which included NPC and other pathologically confirmed non-nasopharyngeal tumours, such as lymphoma, rhabdomyosarcoma, olfactory neuroblastoma, malignant melanoma, and plasmacytoma. A fully convolutional network based on the initial architecture was established to detect nasopharyngeal malignancies. The overall accuracy of the network in the test set reached 88.7%, which was better than that of the experts [124].

#### 6.3.3. 2019

Retrieved none.

#### 6.3.4. 2020

Two similar studies, [125,126], based on pathological images were conducted. The authors in [125] used 1970 whole slide pathological images of 731 cases: 316 cases of inflammation, 138 cases of lymphoid hyperplasia, and 277 cases of NPC. The second study used 726 nasopharyngeal biopsies consisting of 363 images of NPC and 363 of benign nasopharyngeal tissue [126]. In [125], Inception-v3 was used to build the classifier, while ResNeXt, a deep neural network with a residual and inception architecture, was used to build the classifier in [126]. The AUCs obtained in [125,126] were 0.936 and 0.985, respectively.

A study based on 203 NPC and 209 benign nasopharyngeal hyperplasia MRI images to distinguish early NPC from nasopharyngeal benign hyperplasia was conducted [127]. A CNN-based classifier was established, and an AUC of 0.96 and an accuracy of 91.5% were reached, which showed no significant difference for NPC detection when compared to the radiologist (accuracy = 87%).

In [128], 3142 NPC and 958 benign hyperplasia images were used. Of the studies that concentrated on AI tools for NPC imaging, this study was conducted with the largest sample size. A self-constrained 3D DenseNet architecture was developed for tumour detection and segmentation. In the differentiation of NPC and benign hyperplasia, the model showed encouraging performance and obtained higher overall accuracy than that of experienced radiologists (97.77% vs. 95.87%) (Table 9).

### 6.4. Segmentation

Radiotherapy is the most important treatment for NPC. However, it is necessary to accurately delimit the nasopharyngeal tumour volume and the organs at risk in images of the auxiliary damage caused by radiotherapy itself. Therefore, segmentation is particularly relevant to DL in NPC imaging.

#### 6.4.1. 2017

Men [129] developed an end-to-end deep deconvolution neural network (DDNN) to segment tumours, lymph nodes, and risky organs around tumours. A total of 230 patients diagnosed with NPC stages I and II were included. The performance of the DDNN was compared with that of the VGG-16 model. The average DSC value of the DDNN was 80.9% for the total nasopharyngeal tumour volume and 62.3% for the total tumour volume of metastatic lymph nodes, while the DSC values of the VGG-16 were 72.3% and 33.7%, respectively.

#### 6.4.2. 2018

A CNN was used to build an automatic segmentation model for NPC based on enhanced MRIs in Li’s study [130], and case-by-case leave-one-out cross-validation was used to train the network. Their research obtained a DSC value of 0.89. Wang [131] applied a similar method, but only 15 MRI images of patients with NPC were included, and the DSC obtained was 0.79.

Ma’s [132] paper proposed a discriminative learning-based approach for automated NPC segmentation using CT and MRI. The CNN integrated two normal classification sub-networks into a Siamese-like sub-network that could use each other’s multimodal information. The authors concluded that the multimodal method achieves higher segmentation performance (DSC = 0.712) when compared with the segmentation method without multimodal similarity metric learning and the method that only uses CT (DSC = 0.636).

#### 6.4.3. 2019

Daoud [133] proposed a two-stage NPC segmentation method based on CNN using CT images of axial, coronal, and sagittal sections. In the first stage, areas of non-target organs were identified and eliminated from the CT images. The task of the second stage was to identify the NPC from the remaining area of the CT image. The authors concluded that their proposed two-stage segmentation of NPC by integrating three-phase CT images has a satisfactory performance with DSCs of 0.87, 0.85, and 0.91 in axial, coronal, and sagittal sections, respectively.

In [134], a 3D CNN architecture based on VoxResNet was established to automate primary gross tumour volume contouring. It is worth mentioning that this study included a larger sample size (1021 NPCs) than most other studies on this issue. VoxResNet and eight radiation oncologists from multiple centres were evaluated. The DSC of VoxResNet was 0.79, and the accuracies of the oncologists significantly improved with the assistance of VoxResNet (*p* < 0.001).

In the study by Liang [135], a fully automated deep learning method was developed for organs-at-risk detection and segmentation of CT images, and the DSCs for the segmentation of the brain stem, eye lens, larynx, mandible, oral cavity, mastoid, spinal cord, parotid gland, temporomandibular joint, and optic nerve were between 0.689 and 0.934. Zhong [136] conducted a similar study that combined the DL and boosting algorithm to segment the organs at risk, including the parotid gland, thyroid, and optic nerve, and the corresponding DSCs were 0.92, 0.92, and 0.89, respectively.

Ma published another article in 2019 [137] on NPC image segmentation, similar to the study in 2018 [132]. Based on the developed multimodal convolutional neural network (M-CNN), the authors combined the high-level features extracted by the single-mode CNN and M-CNN to form a combined CNN. The study concluded that the model with multi-mode information fusion performs better than the model without the multi-mode information fusion.

Li [138] proposed and trained a U-Net to automatically segment and delineate tumour targets in patients with NPC. A total of 502 patients from a single medical centre were included, and CT images were collected and pre-processed as a dataset. The trained U-Net finally obtained DSCs of 0.659 for lymph nodes and 0.74 for primary tumours in the testing set.

#### 6.4.4. 2020

Xue [139] proposed a sequential and iterative U-Net (SI-Net) to automatically segment the target volume of the primary tumour and compared it with a conventional U-Net. It was considered that the performance of the SI-Net was better than that of the U-Net (DSCs were 0.84 and 0.80, respectively).

Chen [140] proposed a novel multimodal MRI fusion network to segment NPCs accurately using T1, T2, and contrast-enhanced T1 MRI. The network model was composed of a 3D convolutional block attention module and a residual fusion block and adopted a self-transfer training strategy. A total of 149 patients with NPC were included. The network model obtained a DSC value of 0.724.

In [141], a 3D CNN with a long-range skip connection and multi-scale feature pyramid was developed for NPC segmentation. The network was trained and tested on the 3D MRI images of 120 patients with NPC using five-fold cross-validation, and the 3D CNN achieved a DSC of 0.737.

Ye [142] developed a CNN model based on dense connectivity embedding U-Net to automatically segment primary tumours of NPC on a dual-sequence MRI. A total of 44 MRI images of patients with NPC were included in this study. The average DSC of the external subjects, which consisted of seven patients with NPC, was 0.87.

Considering that NPC is a malignant tumour with a tendency to invade the surrounding tissues, in a complex MRI background, it is difficult to distinguish the signs of invasion on the edge from the closely connected normal tissues. To address the background dominant problem in improving the segmentation accuracy of NPC, Li [143] proposed a coarse-to-fine deep neural network, which started by predicting a coarse mask based on a well-designed segmentation module, followed by a boundary rendering module, which exploited semantic information from different layers of feature maps to refine the boundary of the coarse mask. The dataset contained 2000 MRI slices from 596 patients, and the DSC of the model was 0.703.

Jin [144] proposed a ResSE-UNet network with a ternary cross-entropy loss function to segment the total volume of the primary tumour and compared it with the tumour segmentation model based on the original U-Net, U-Net-NN. The data set of the study consisted of 1757 CT slices from 90 patients with NPC, and ResSE-UNet obtained the best DSC (0.84).

In [145], Wang proposed a modified version of the 3D U-Net model with Res-blocks and SE-blocks to segment the gross tumour volume of the nasopharynx. The novelty of the research is that an automatic pre-processing method was proposed to crop the 3D region of interest of the nasopharynx gross tumour volume, which improved the efficiency of image pre-processing. Automatic delineation models based on 3D U-Net, 3DCNN, and 2D DDNN were developed. The DSCs of the three models were 82.70%, 80.54%, and 77.97%, respectively.

Ke [128] developed a self-constrained 3D DenseNet architecture for tumour detection and segmentation, which was described in the NPC detection and/or diagnosis section. In terms of automatic segmentation of the tumour area, the architecture obtained a good performance with a DSC of 0.77 in the test cohort.

#### 6.4.5. 2021

CNN shows promise for segmenting malignant tumours on contrast-enhanced MRIs, but there are situations where contrast agents are not suitable for specific patients. Can CNN accurately segment tumours based on MRI images without enhancement? To clarify this issue, Wong [146] developed a U-Net to segment primary NPC on a non-contrast-enhanced MRI and compared it with a contrast-enhanced MRI. The U-Net suggested a similar performance (DSC = 0.71) between fat suppression (fs)-T2W and enhanced-T1W, and the enhanced-fs-T1W images showed the highest DSC (0.73).

Bai [147] fine-tuned a pre-trained ResNeXt-50 U-Net, which uses the recall preserved loss to produce a rough segmentation of the gross tumour volume of NPC. Then, the well-trained ResNeXt-50 U-Net was applied to the fine-grained gross tumour volume boundary minute. The study obtained a DSC of 0.618 for online testing (Table 10).

## 7. Deep Learning-Based Radiomics

DL has shown great potential to dominate the field of image analysis. In ROI [148] and feature extraction tasks [149,150], which lay in the implementation pipeline of radiomics, DL has achieved good results. After completing the model training, DL can automatically analyse images, which is one of the greatest strengths compared to radiomics. Many researchers have introduced DL into radiomics (termed deep learning-based radiomics, DLR) and achieved encouraging results [151]. This may be a trend for the application of AI tools in medical imaging in the future. Therefore, we list these studies on NPC imaging separately.

### 7.1. Studies Based on Deep Learning-Based Radiomics (DLR)

#### 7.1.1. 2017

Retrieved none.

#### 7.1.2. 2018

To investigate the feasibility of radiomics for the analysis of radioresistance, Li [152] trained and validated an artificial neural network, a k-nearest neighbour, and an SVM model using stratified ten-fold cross-validation. Pre-processed MRI images were used for feature extraction, and principal component analysis was performed for feature reduction. The author concluded that radiomic analysis can be served as imaging biomarkers to facilitate early salvage for patients with NPC who are at risk of in-field recurrence. However, only 20 patients with recurrent NPC were recruited for this study.

#### 7.1.3. 2019

Peng [30] developed a model based on DLR to predict the effect of induction chemotherapy on patients with advanced NPC. The research constructed and trained four deep CNN models to extract the features in the ROI, which alternated the feature extraction step of radiomics. LASSO was used to screen the features, and a nomogram that integrated clinical indicators was developed. Finally, the study reported a C-index of 0.722 in the test set.

#### 7.1.4. 2020

Similar to the study of Peng [30], Zhong [153] established a radiomic nomogram based on MRI images and clinical features and adopted a DCNN (SE-ResNeXt) to quantify the tumour phenotype end-to-end in the process of image feature extraction. The established radiomic nomogram obtained a C-index of 0.788 in the test cohort.

In [154], Zhang innovatively combined the clinical features of patients with nasopharyngeal cancer, the radiomic features based on MRIs, and the DCNN model based on pathological images to construct a multi-scale nomogram to predict the failure-free survival of patients with NPC. The nomogram showed a consistent significant improvement for predicting treatment failure compared with the clinical model in the internal test (C-index: 0.828 vs. 0.602, *p* < 0.050) and external test (C-index: 0.834 vs. 0.679, *p* < 0.050) cohorts. (Table 11)

## 8. Discussion

The widespread application of AI tools in the medical field is a promising trend in the future of medicine. Radiomics and artificial neural networks could be the main approaches to achieve this and also be valuable tools to completely change the strategy of clinical diagnosis and treatment of tumours [28,155]. Particularly in brain [156,157], breast [158,159], lung [160,161], and prostate tumours [162,163], the application of radiomics and DL is at the forefront and shows great potential [164]. Although radiomics has been applied to medical imaging since 2012 and DL has started to be applied to medical imaging around 2015, their application in NPC has only gradually begun in 2017. Many attempts have been made to apply AI tools for NPC imaging in clinical settings. However, there are still some limitations in this field.

First, the current high-quality research in these areas is insufficient. For example, the current research is based on cross-sectional images, such as the most common use of pre-treatment images for prediction. Research based on time-varying images has not yet been conducted. The treatment of most tumours, especially NPC, is a long and multi-cycle combined process. It is unrealistic to guide the entire treatment process based only on the tumour images before treatment. Therefore, it is of great value to evaluate the real-time response of tumours to drugs and the radiation injury risk of important organs around the tumour based on the dynamic changes of image features during treatment. This can provide clinicians with key information to optimize treatment strategies. Furthermore, the quality and size of the dataset used for model training are extremely critical and are the limit of the accuracy and generalization ability of the established model. However, the number of cases included in most studies is limited, and many studies have not performed external testing of the model. In a recent systematic review [165], studies on MRI-based radiology in NPC published in recent years were evaluated using their radiomics quality scores. It was found that only 8% of the included studies included external validation, and the author concluded that radiomics articles in NPC were mostly of low methodological quality. This reflects the current, frustrating situation in this field. Moreover, owing to the lack of massive amounts of structured data, it is difficult for most algorithms to be implemented in clinical practice [166], which is one of the most urgent problems to be solved in the future development of AI tools for NPC.

Second, there are still some aspects that have not been covered in NPC imaging using radiomics and DL. For example, the newly proposed radiogenomic approach combines radiology and genomics [167,168]. Radiogenomics hypothesizes that the texture heterogeneity of tumours could reflect the difference in quantitative imaging features between genome and molecular phenotype, which could indicate the subtype, prognosis, drug response, and other information of the tumour [169]. It has experienced tremendous growth in studies of gliomas [170], breast cancer [171,172], colorectal cancer [173], lung cancer [174], and ovarian cancer [175] over the past years. Radiogenomics has demonstrated significant potential for developing non-invasive prognostic and diagnostic methods, identifying biomarkers for treatment, tumour phenotyping, and genomic signatures [176]. Precision medicine is a disease treatment and prevention approach that considers individual differences in genes, environment, and lifestyle and integrates multiple sources of information to achieve the ultimate goal of personalized management [177]. Radiogenomics, which bridges imaging and genomics [178], could provide a new, non-invasive, fast, and low-cost approach for the implementation of precision medicine [179]. However, to the best of our knowledge, studies on radiogenomics for NPC are currently lacking.

Radiomics is based on feature engineering and retains an inherent defect that manually delineating the area of interest is strictly required, which is labour-consuming. DL could provide a solution for this problem [25]; in addition, it can provide an efficient method for feature extraction in the radiomics pipeline [180]. Therefore, DLR, which combines the advantages of DL and radiomics, has been proposed and widely researched [25,181,182]. Although there are four studies that use DLR in NPC [30,152,153,154], the method is far from being fully developed.

The purpose of AI tools in NPC imaging is to be used in clinical practice. However, there are still many limitations and gaps between research and clinical applications. The lack of massive structured data is the most urgent problem to be solved. Considering that one of the biggest challenges for oncology is to develop accurate and cost-effective screening procedures [28], fast and minimal manual work will be a common clinical need in the future. Therefore, excessive human intervention in the use of developed AI tools is a block that must be handled. For example, the time-consuming step of segmentation and manual selection of image layers based on experience to construct a dataset with the most obvious tumour images that were expected to perform better. From this point, we can perceive the signs of deep learning by replacing traditional radiomics. However, it is difficult for the huge-data-based DL, which is capable of fully automated analysis, to obtain sufficient labelled data to develop state-of-the-art models and eliminate radiomics in the near future, considering that the research field of radiomics is mostly composed of topics such as prognosis of tumours and the efficacy of tumours on treatments, which are costly and time-consuming for data collection. Introducing DL into the pipeline of radiomics to improve the accuracy and stability of the established model may be a promising method in the short term.

After summarizing the relevant studies on deep learning and radiomics for NPC imaging, it was found that MRI was adopted by most studies as established datasets to carry out tasks, such as segmentation of lesions and tissues at risk, generation of synthetic high-quality CT images, and radiotherapy planning. This may be due to the better resolution of soft tissue by MRI than CT. However, different types of medical image have different advantages in machine learning. Therefore, which images are used to establish the dataset should be determined according to the specific task at hand. For example, CT has a higher definition than MRI in showing the damage of NPC to the skull base bone. Therefore, when conducting skull base bone-related research, CT-based models may have better performance. The endoscopic image is a special feature of nasopharyngeal carcinomas which is different from most tumours. The endoscopic image of the nasopharynx is of great significance for the early screening of nasopharyngeal carcinoma, which is difficult to achieve by MRI and CT. However, there are only a few studies in this area based on endoscopic images. Pathological slices are used in many studies related to deep learning and tumours, but there are only a few studies related to NPC. In fact, considerable work remains to be done in this area, such as automatic reporting of specific pathological types of nasopharyngeal cancer and immunohistochemical results, automated predictions, slice-based prognostic analysis of patients, etc. Therefore, it is necessary to adopt datasets containing corresponding medical images according to the specific task.

It is worth mentioning that, from the clinician’s perspective, we hope that all relevant studies have the potential to promote the application of AI tools in clinical practice. However, the clinical significance of studies based on AI tools for predicting T staging of tumours is discounted. The TNM staging system, which is essentially a prediction system for tumour prognosis, represents a body of knowledge combining evidence-based findings from clinical studies with empirical knowledge from site-specific experts [183,184]. The TNM staging system is composed of the relationship between the tumour and the normal anatomical structure around it, which only presents a small amount of information on the image that could be recognized by the naked eye. More importantly, using AI tools to predict tumour prognosis will challenge the core of the TNM staging system in the future, because they have a parallel relationship (Figure 6) and most of the prognostic prediction models based on radiomics or DL are already better than the TNM staging system [24,34,56,57]. Therefore, there are limitations to using AI to predict T staging to solve clinical problems.

## 9. Conclusions and Future Work

In this review, the studies of NPC image analysis based on radiomics and DL after 2017 are comprehensively summarized (Figure 7). Before summarizing these studies, we provide a brief description of radiomics and DL. We then divided the studies into three categories: radiomics-, DL-, and DLR-based, according to the methods adopted in the studies. The radiomics-based studies were divided into five categories: prognosis prediction, assessment of tumour metastasis, diagnosis, prediction of therapeutic effect, and predicting complications and were summarized in chronological order. The DL-based studies were divided into four categories: prognosis prediction, image synthesis, detection and/or diagnosis, and segmentation and were summarized in chronological order. Due to the limited number of studies, we summarize the DLR-based research in chronological order. According to this method, we present a full picture of the application of radiomics and DL in NPC imaging.

Research on radiomics and DL in NPC imaging has only started in recent years. Therefore, there are still many issues that need further research in the future: linking NPC imaging features with tumour genes/molecules to promote the development of precision medicine for non-invasive, rapid, and low-cost approaches; using multi-stage dynamic imaging to assess tumour response to drugs/radiotherapy and predict the risk of radiation therapy in surrounding vital organs to guide treatment decisions; and bridging the gap from the AI tools established in studies to clinical applications. In addition, current studies based on nasal endoscopic images and pathological images are lacking. In particular, accurate and rapid screening of NPC is of great significance, considering that endoscopic images are usually the primary screening images for most patients. Further high-quality research in this regard is needed. Finally, there is still a lack of large-scale, comprehensive, and fully labelled datasets for NPC; datasets similar to those that are available for lung and brain tumours. The establishment of large-scale public datasets is an important task in the future.

## Figures and Tables

**Figure 1 diagnostics-11-01523-f001:**
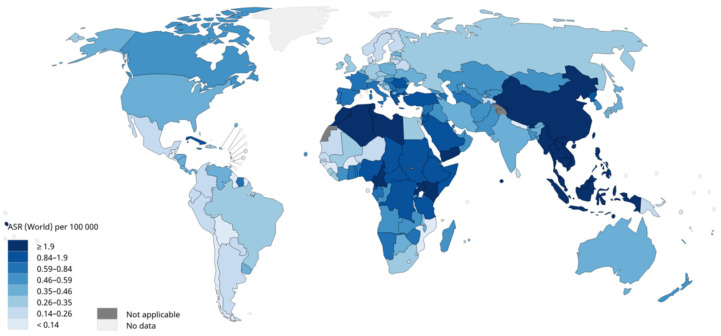
Estimated age-standardised incidence rates (World) in 2020, nasopharynx, both sexes, all ages. ARS: age-standardised rates. Data source: GLOBOCAN 2020. Graph production: IARC (http://gco.iarc.fr/today, accessed on 31 March 2021.) World Health [2].

**Figure 2 diagnostics-11-01523-f002:**
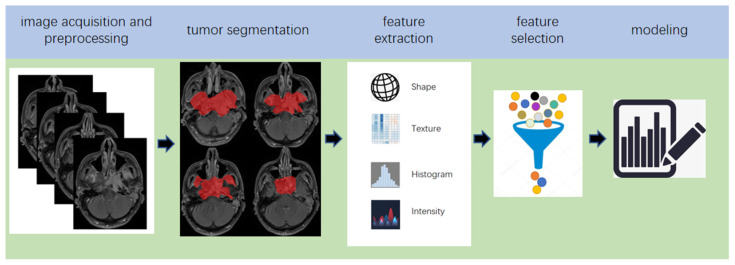
Five steps of the pipeline of radiomics.

**Figure 3 diagnostics-11-01523-f003:**
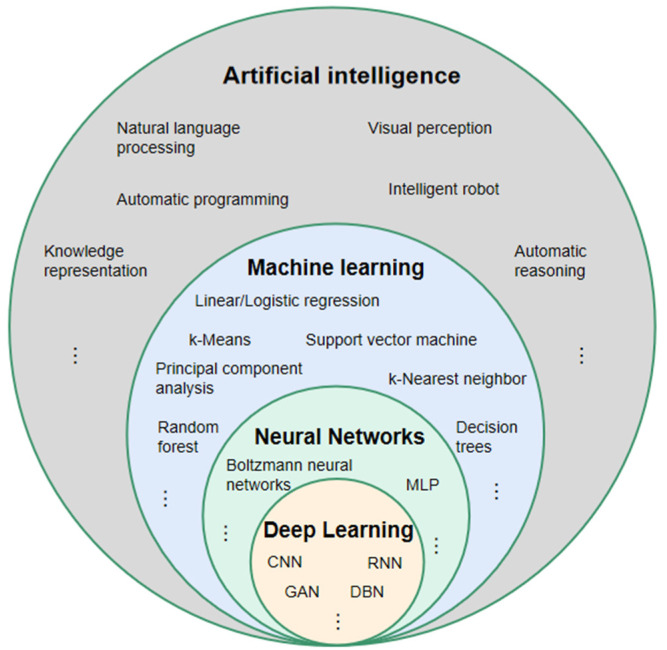
Relationship between artificial intelligence, machine learning, neural network, and deep learning. MLP: multilayer perception; CNN: convolutional neural network; RNN: recurrent neural network; DBN: deep belief network; GAN: generative adversative network.

**Figure 4 diagnostics-11-01523-f004:**
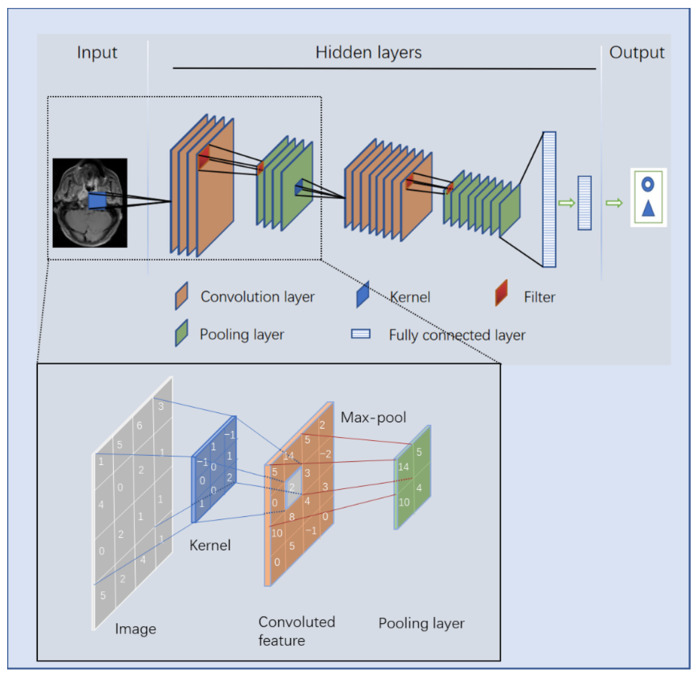
Image processing principle of a convolutional neural network.

**Figure 5 diagnostics-11-01523-f005:**
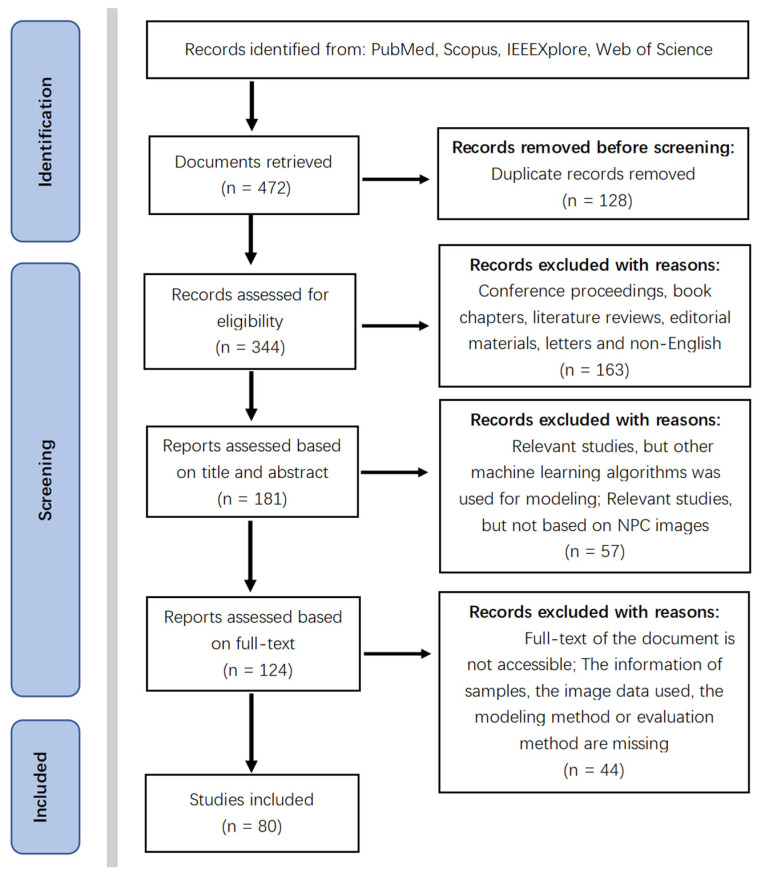
PRISMA flow diagram of the search result.

**Figure 6 diagnostics-11-01523-f006:**
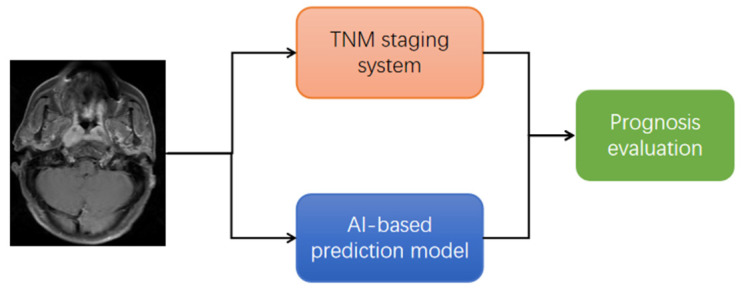
Parallel relationship between TNM staging system and artificial intelligence (AI)-based prediction model.

**Figure 7 diagnostics-11-01523-f007:**
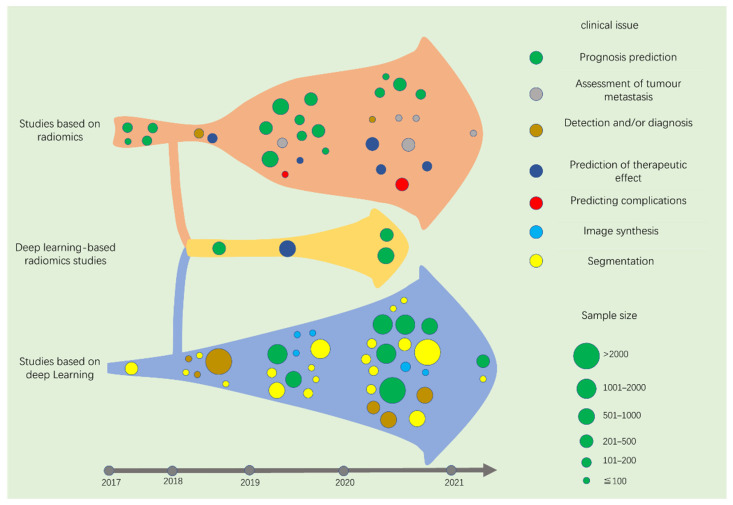
The distribution of studies based on radiomics, deep learning, and deep learning-based radiomics from 2017 to early 2021.

**Table 1 diagnostics-11-01523-t001:** Inclusion and exclusion criteria of the study.

Criterias	Detailed Rules and Regulations
Inclusion Criteria	Journal articles published in the English language; Studies published from 2012 to date; Original researches; Full-text papers that are accessible; Radiomics was used to analyze the images of NPC; Deep learning was used to analyze the images of NPC; The samples, the image data used, the modeling method and evaluation method are described in detail.
Exclusion Criteria	Papers that are written in other languages; Full-text of the document is not accessible on the internet; Relevant studies, but other machine learning algorithms that are not based on deep learning or radiomics were used for modeling; Relevant studies, but not based on NPC images; The information of samples, the image data used, the modeling method or evaluation method are not described; Conferences papers, literature reviews, and editorial materials that do not belong to original researches.

**Table 2 diagnostics-11-01523-t002:** Studies of predicting the prognosis of nasopharyngeal carcinoma (NPC) using radiomics.

Author, Year, Reference	Image	Sample Size (Patient)	Feature Selection	Modeling	Model Evaluation
Zhang, B. (2017) [24]	MRI	108	LASSO	CR, nomograms, calibration curves	C-index 0.776
Zhang, B. (2017) [81]	MRI	110	L1-LOG, L1-SVM, RF, DC, EN-LOG, SIS	L2-LOG, KSVM, AdaBoost, LSVM, RF, Nnet, KNN, LDA, NB	AUC 0.846
Zhang, B. (2017) [82]	MRI	113	LASSO	RS	AUC 0.886
Ouyang, F.S. (2017) [83]	MRI	100	LASSO	RS	HR 7.28
Lv, W. (2019) [84]	PET/CT	128	Univariate analysis with FDR, SC > 0.8	CR	C-index 0.77
Zhuo, E.H. (2019) [85]	MRI	658	Entropy-based consensus clustering method	SVM	C-index 0.814
Zhang, L.L. (2019) [86]	MRI	737	RFE	CR and nomogram	C-index 0.73
Yang, K. (2019) [87]	MRI	224	LASSO	CR and nomogram	C-index 0.811
Ming, X. (2019) [88]	MRI	303	Non-negative matrix factorization	Chi-squared test, nomogram	C-index 0.845
Zhang, L. (2019) [89]	MRI	140	LR-RFE	CR and nomogram	C-index 0.74
Mao, J. (2019) [90]	MRI	79	Univariate analyses	CR	AUC 0.825
Du, R. (2019) [91]	MRI	277	Hierarchal clustering analysis, PR	SVM	AUC 0.8
Xu, H. (2020) [92]	PET/CT	128	Univariate CR, PR > 0.8	CR	C-index 0.69
Shen, H. (2020) [93]	MRI	327	LASSO, RFE	CR, RS	C-index 0.874
Bologna, M. (2020) [94]	MRI	136	Intra-class correlation coefficient, SCC > 0.85	CR	C-index 0.72
Feng, Q. (2020) [95]	PET/MR	100	LASSO	CR	AUC 0.85
Peng, L. (2021) [96]	PET/CT	85	W-test, Chi-square test, PR, RA	SFFS coupled with SVM	AUC 0.829

Least absolute shrinkage and selection operator (LASSO), L1-logistic regression (L1-LOG), L1-support vector machine (L1-SVM), random forest (RF), distance correlation (DC), elastic net logistic regression (EN-LOG), sure independence screening (SIS), L2-logistic regression (L2-LOG), kernel support vector machine (KSVM), linear-SVM (LSVM), adaptive boosting (AdaBoost), neural network (Nnet), K-nearest neighbour (KNN), linear discriminant analysis (LDA), and naive Bayes (NB).

**Table 3 diagnostics-11-01523-t003:** Studies for assessing tumour metastasis using radiomics.

Author, Year, Reference	Image	Sample Size	Feature Selection	Modeling	Model Evaluation
Zhang, L. (2019) [97]	MRI	176	LASSO	LR	AUC 0.792
Zhong, X. (2020) [98]	MRI	46	LASSO	Nomogram	AUC 0.72
Akram, F. (2020) [99]	MRI	14	Paired t-test and W-test	Shapiro-Wilk normality tests	*p* < 0.001
Zhang, X. (2020) [100]	MRI	238	MRMR combined with 0.632 + bootstrap algorithms	RF	AUC 0.845
Peng, L. (2021) [96]	PET/CT	85	W-test, PR, RA, Chi-square test	SFFS coupled with SVM	AUC 0.829

Maximum relevance minimum redundancy (MRMR).

**Table 4 diagnostics-11-01523-t004:** Studies of nasopharyngeal carcinoma (NPC) diagnosis using radiomics.

Author, Year, Reference	Image	Sample Size	Feature Selection	Modeling	Model Evaluation
Lv, W. (2018) [101]	PET/CT	106	Intra-class coefficient	LR with LOOCV	AUC 0.89
Du, D. (2020) [102]	PET/CT	76	MIM, FSCR, RELF-F, MRMR, CMIM, JMI, SC > 0.7	DT, KNN, LDA, LR, NB, RF, and SVM with radial basis function kernel	AUC 0.892

Logistic regression with leave-one-out cross-validation (LOOCV), mutual information maximization (MIM), Relief-F (RELF-F), conditional mutual information maximization (CMIM), Fisher score (FSCR), joint mutual information (JMI), Spearman’s correlation (SC).

**Table 5 diagnostics-11-01523-t005:** Studies for the prediction of therapeutic effect of nasopharyngeal carcinoma (NPC) using radiomics.

Author, Year, Reference	Image	Sample Size	Feature Selection	Modeling	Model Evaluation
Wang, G. (2018) [103]	MRI	120	LASSO	LR	AUC 0.822
Yu, T.T. (2019) [104]	MRI	70	LASSO	Univariate LR	AUC 0.852
Yongfeng, P. (2020) [105]	MRI	108	ANOVA/MW test, correlation analysis and LASSO	Multivariate LR	AUC 0.905
Zhang, L. (2020) [106]	MRI	265	LASSO	LR	AUC 0.886, 0.863
Zhao, L. (2020) [107]	MRI	123	t-test and LASSO based on LOOCV	SVM, nomogram, backward stepwise LR	C-index 0.863

**Table 6 diagnostics-11-01523-t006:** Studies for predicting the complications of radiation therapy using radiomics.

Author, Year, Reference	Image	Sample Size	Feature Selection	Modeling	Model Evaluation
Liu, Y. (2019) [108]	CT	35	RFE	LR	Precision 0.922
Zhang, B. (2020) [109]	MRI	242	The relief algorithm	RF	AUC 0.830

Recursive feature elimination (RFE).

**Table 7 diagnostics-11-01523-t007:** Studies for the prognosis prediction of nasopharyngeal carcinoma (NPC) based on deep learning (DL).

Author, Year, Reference	Image	Sample Size	Modeling	Model Evaluation
Qiang, M.Y. (2019) [110]	MRI	1636	3D DenseNet	HR 0.62
Du, R. (2019) [111]	MRI	596	DCNN	AUC 0.69
Yang, Q. (2020) [112]	MRI	1138	Resnet network	AUC 0.943
Jing, B. (2020) [113]	MRI	1417	Multi-modality deep survival network	C-index 0.651
Qiang, M. (2020) [34]	MRI	3444	3D-CNN	C-index 0.776
Cui, C. (2020) [114]	MRI	792	Automatic machine learning (AutoML) including DL	AUC 0.796
Liu, K. (2020) [115]	Pathology	1055	Neural network DeepSurv	C-index 0.723
Zhang, L. (2021) [116]	MRI	233	Resnet network	AUC 0.808

**Table 8 diagnostics-11-01523-t008:** Studies for image synthesis of nasopharyngeal carcinoma (NPC) based on deep learning (DL).

Author, Year, Reference	Image	Sample Size	Modeling	Model Evaluation
Li, Y. (2019) [117]	CBCT	70	U-Net neural network (DCNN)	1%/1 mm GPR 95.5%
Wang, Y. (2019) [118]	MRI	33	U-Net neural network (DCNN)	MAE: 97 ± 13 HU in soft tissue, 131 ± 24 HU in all region, 357 ± 44 HU in bone
Tie, X. (2020) [119]	MRI	32	ResU-Net	Structural similarity index 0.92
Peng, Y. (2020) [120]	MRI	173	GANs	2%/2mm GPR 98.52~98.68%

Generative adversarial networks (GANs).

**Table 9 diagnostics-11-01523-t009:** Studies of nasopharyngeal carcinoma (NPC) detection and/or diagnosis based on deep learning (DL).

Author, Year, Reference	Image	Sample Size	Modeling	Model Evaluation
Mohammed, M.A. (2018) [121,122,123]	Endoscopic images	381 images	ANN	Accuracy 96.22%
Li, C. (2018) [124]	Endoscopic images	28,966 images	Fully convolutional network (FCNN)	Accuracy 88.7%
Diao, S. (2020) [125]	Pathology	731 patients	Inception-v3	AUC 0.936
Chuang, W.Y. (2020) [126]	Pathology	726 patients	ResNeXt	AUC 0.985
Wong, L.M. (2020) [127]	MRI	412 patients	Residual Attention Network (RAN)	AUC 0.96
Ke, L. (2020) [128]	MRI	4100 patients	3D DenseNet	Accuracy 97.77%

**Table 10 diagnostics-11-01523-t010:** Studies for nasopharyngeal carcinoma (NPC) segmentation based on deep learning (DL).

Author, Year, Reference	Image	Sample Size	Modeling	Model Evaluation
Men, K. (2017) [129]	CT	230	DDNN and VGG-16	DSC GTVnx 80.9% GTVnd 62.3% CTV 82.6%
Li, Q. (2018) [130]	MRI	29	CNN	DSC 0.89
Wang, Y. (2018) [131]	MRI	15	DCNN	DSC 0.79
Ma, Z. (2018) [132]	CT, MRI	50	Multi-modality CNN	DSC 0.636 (CT), 0.712 (MRI)
Daoud, B. (2019) [133]	CT	70	CNN	DSC 0.91
Lin, L. (2019) [134]	MRI	1021	3D-CNN	DSC 0.79
Liang, S. (2019) [135]	CT	185	CNN	DSC 0.689–0.937
Zhong, T. (2019) [136]	CT	140	CNN	DSC Parotids 0.92 Thyroids 0.92 Optic nerves 0.89
Ma, Z. (2019) [137]	CT, MRI	90	Single-modality CNN multi-modality CNN	DSC 0.746 (CT), 0.752 (MRI)
Li, S. (2019) [138]	CT	502	U-Net (CNN)	DSC Lymph nodes 0.659 Tumor 0.74
Xue, X. (2020) [139]	CT	150	SI-Net and U-Net	DSC 0.84
Chen, H. (2020) [140]	MRI	149	3D-CNN	DSC 0.724
Guo, F. (2020) [141]	MRI	120	3D-CNN	DSC 0.737
Ye, Y. (2020) [142]	MRI	44	Dense connectivity embedding U-net	DSC 0.87
Li, Y. (2020) [143]	MRI	596	ResNet-101	DSC 0.703
Jin, Z. (2020) [144]	CT	90	ResSE-UNet	DSC 0.84
Wang, X. (2020) [145]	CT	205	3D U-Net	DSC 0.827
Ke, L. (2020) [128]	MRI	4100	3D DenseNet	DSC 0.77
Wong, L.M. (2021) [146]	MRI	195	CNN	DSC 0.73
Bai, X. (2021) [147]	MRI	60	ResNeXt-50 and U-Net	DSC 0.618

**Table 11 diagnostics-11-01523-t011:** Studies of nasopharyngeal carcinoma (NPC) imaging based on deep learning-based radiomics (DLR).

Author, Year, Reference	Image	Sample Size	Feature Selection	Modeling	Model Evaluation
Li, S. (2018) [152]	CT	306	ICC, PCC, and PCA	ANN, KNN, SVM	AUC ANN: 0.812 KNN: 0.775 SVM: 0.732
Peng, H. (2019) [30]	PET/CT	707	LASSO	DCNNs and nomogram	C-index 0.722
Zhong, L.Z. (2020) [153]	MRI	638	Not described	SE-ResNeXt, CR and nomogram	C-index 0.788
Zhang, F. (2020) [154]	MRI, Pathology	220	ICC > 0.75, Univariate analysis, MRMR, RF	ResNet-18, Nomogram	C-index 0.834

Intraclass correlation coefficients (ICC), Pearson correlation coefficient (PCC), and principal component analysis (PCA).

## Data Availability

Data sharing is not applicable to this article.

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
