# Peer review of "A Comprehensive Review on Radiomics and Deep Learning for Nasopharyngeal Carcinoma Imaging"

_diagnostics, 2021, doi:10.3390/diagnostics11091523_

Round 1
Reviewer 1 Report
This is a well written review article that comprehensively summarized the progress in the application of AI-based image analysis for management of Nasopharyngeal carcinoma (NPC). Many studies were reviewed, and for each study, the subjects, methods, and major findings were described. The article was well structured , first briefly describing the methodology of the two AI methods: radiomics and deep learning (DL), and then summarizing the published studies for different clinical tasks. For radiomics studies, they were separated into 5 categories: prognosis prediction, assessment of tumour metastasis, tumour diagnosis, prediction of therapeutic effect, and prediction of complications. For DL, they were divided into 4 categories: prognosis prediction, image synthesis, detection and/or diagnosis, and image segmentation. Then, a newer method using combined DL and radiomics was proposed, which has the potential for detection of NPC in screening examinations. Overall, this is an excellent review article with informative summary of published studies and a good future direction. I just had two suggestions:
- Since the focus is for management of NPC based on image analysis, it will be great to add a paragraph talking about the current use of different imaging examinations for NPC management, and the need of AI-based analysis methods. For example, 1) what are the roles of MRI, CT, PET/CT, endoscopic imaging, and pathological slides; 2) how they are used in detection, diagnosis, radiotherapy RT planning (such as segmentation of lesion and organs-at-risk, and generation of synthetic high-quality CT images), treatment response evaluation, and prognosis prediction; and then 3) why the AI-based image analysis is needed.
- Please talk about the respective strengths and drawbacks of radiomics and DL to guide readers in their different clinical applications. For example, radiomics is usually based on the segmented tissues, while it’s time consuming, yet can directly extract the contained features in the lesion related to the clinical classification (e.g., benign vs. malignant, more vs. less aggressive, etc.). In contrast, for diagnosis or prediction of aggressiveness using DL, it does not need a precise segmentation, and usually a region containing the abnormality is used as input, which can simplify the pre-processing procedure; yet the input contains information from non-lesion tissues and often results in lower performance. However, for therapy planning, DL provides a suitable tool for lesion/organ segmentation and generation of synthetic images. It would be great to add a paragraph talking about these before going into details in sections 5, 6, 7 – so the readers can follow the summarized studies with a better understanding of the rationale behind them. In can be added into Section 3 or after the description of screening studies in Section 4.
Author Response
- Since the focus is for management of NPC based on image analysis, it will be great to add a paragraph talking about the current use of different imaging examinations for NPC management, and the need of AI-based analysis methods. For example, 1) what are the roles of MRI, CT, PET/CT, endoscopic imaging, and pathological slides; 2) how they are used in detection, diagnosis, radiotherapy RT planning (such as segmentation of lesion and organs-at-risk, and generation of synthetic high-quality CT images), treatment response evaluation, and prognosis prediction; and then 3) why the AI-based image analysis is needed.
Response:
Thank you very much for your valuable suggestions. We have discussed how to select medical images (MRI, CT, endoscopic imaging, and pathological slides) as the dataset of AI tasks and the advantages of different medical images (a new paragraph was added to the discussion).
- Please talk about the respective strengths and drawbacks of radiomics and DL to guide readers in their different clinical applications.....
Response:
Thank you for your suggestion. The shortcomings of radiomics and deep learning are partially discussed in the fifth paragraph of section 8 (lines 764-774), although not comprehensive. Your suggestion is also very valuable, and we have added the analysis of the advantages and disadvantages of radiomics and deep learning in section 3 as a summary of the previous content for the readers following the summarized studies with a better understanding of the rationale behind them.
Reviewer 2 Report
This was an interesting read. I only have some minor comments.
Authors should consider shortening the following sections:
- Pipeline of Radiomics
- The Principle of DL
Other papers have been published investigating the same issue (e.g. doi: 10.1016/j.ejrad.2021.109744); please discuss.
Please, revise the English Language throughout the manuscript.
Author Response
- Authors should consider shortening the following sections: Pipeline of Radiomics, the Principle of DL
Response:
Thank you very much for your suggestion. The pipeline of Radiology and the principle of DL have reached about 800 words respectively. We have carefully considered whether we should make the statement more concise. However, considering the following two reasons, we did not further simplify it. Firstly, this review is for clinicians who do not grasp the principles of AI, so we believe it is necessary to describe the relevant principles more carefully; Secondly, considering the large length of this review, simplifying the length of the pipeline of radiomics and the principle of DL may lead to an imbalance in the content of the manuscript.
- Other papers have been published investigating the same issue (e.g. DOI: 10.1016/j. ejrad.2021.109744); please discuss.
Response:
Thank you very much for your recommendation of the newly published review that related to us in time, which provides us with a valuable reference. Because our review was written in May, we have not previously retrieved the review you recommended. This review systematically evaluates the methodological quality of studies using MRI radiomics for nasopharyngeal cancer patient evaluation and concludes that the greatest limitations are the lack of external validation, biological correlates, prospective design, and open science. We can't agree more on this and we indeed mentioned this worrying situation in the second paragraph of the discussion section (original: "However, the number of cases included in most studies is limited, and many studies have not performed external testing of the model."), despite that, we did not make a quantitative analysis of this phenomenon as this review did. According to your suggestion, we have discussed this review in the second paragraph of the discussion section.
- Please, revise the English Language throughout the manuscript.
Response:
Thank you for your suggestion. We have sought the help of native English professionals to polish the manuscript before submission. In this revision, we have carefully reviewed the English language.